# Curriculum-based Co-design of Morphology and Control of Voxel-based Soft Robots

**Yuxing Wang[1,2*], Shuang Wu[2], Haobo Fu[2†], Qiang Fu[2], Tiantian Zhang[1], Yongzhe Chang[1], Xueqian Wang[1†]**

[1]Tsinghua Shenzhen International Graduate School, Tsinghua University, Shenzhen, China
[2]Tencent AI Lab, Shenzhen, China

## ABSTRACT

Co-design of morphology and control of a Voxel-based Soft Robot (VSR) is challenging due to the notorious bi-level optimization. In this paper, we present a Curriculum-based Co-design (CuCo) method for learning to design and control VSRs through an easy-to-difficult process. Specifically, we expand the design space from a small size to the target size gradually through a predefined curriculum. At each learning stage of the curriculum, we use reinforcement learning to simultaneously train the design policy and the control policy, which is enabled by incorporating the design process into the environment and using differentiable policy representations. The converged morphology and the learned policies from last stage are inherited and then serve as the starting point for the next stage. In empirical studies, we show that CuCo is more efficient in creating larger robots with better performance by reusing the practical design and control patterns learned within each stage, in comparison to prior approaches that learn from scratch in the space of target size.

## 1 INTRODUCTION

The philosophy of embodied cognition (Pfeifer & Bongard, 2006; Pfeifer et al., 2014) inspires the domain of robotics that a robot's ability to interact with the environment depends both on its brain (control policy) and body (morphology), which are inherently coupled (Spielberg et al., 2019; Gupta et al., 2021). However, finding an optimal robot morphology and its controller for solving a given task is often unfeasible. The major challenge for this endeavor is the enormous combined design and policy space. Firstly, the freedom to pick the number of multi-material modules and the ways they are connected makes it notoriously difficult to explore the design space (Medvet et al., 2022). For instance, in a robot simulator (Liu et al., 2020), there are over $4 \times 10^8$ possible morphologies for a robot composed of only 12 modules. Secondly, the evaluation of a morphology requires a separate training procedure for its unique controller.

In this work, we consider the co-optimization of design and control of Voxel-based Soft Robots (VSRs) (Bhatia et al., 2021), a form of modular soft robots composed of elastic, multi-material cubic blocks. Unlike fragile fully-integrated robots, they can be easily disassembled and reassembled to adapt to a wide range of environments (Shah et al., 2020; Pigozzi et al., 2022). For efficiently exploring the modular robot design space, prior approaches commonly rely on artificial evolution (Sims, 1994; Cheney et al., 2013; Medvet et al., 2021), which maintains a population of design prototypes and adopts a bi-level optimization loop, where the outer loop of morphology optimization is based on the fitness of individual controllers from the inner loop. These methods, however, tend to learn from scratch in the target design space where there is a significant combinatorial explosion. Thus, they spend a large amount of time on policy optimization and evaluation. Additionally, their separate training procedures significantly hinder the experience of design and control to be shared across different robots.

---

*This work was done during an internship at Tencent AI Lab, Shenzhen, China.
†Correspondence to: Haobo Fu (haobofu@tencent.com) and Xueqian Wang (wang.xq@sz.tsinghua.edu.cn)

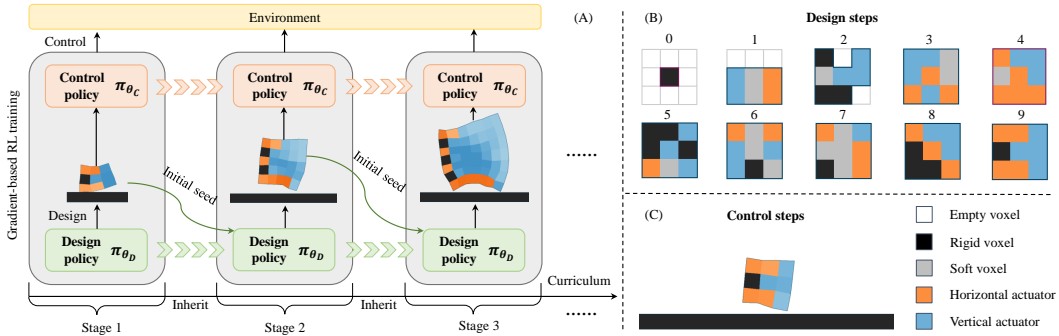

Figure 1: Schematic of our approach CuCo. **(A)** We investigate the brain-body co-design problem of VSRs by establishing an easy-to-difficult optimization process between a small design space (e.g., $3 \times 3$) to the target space (e.g., $7 \times 7$). **(B)** In $3 \times 3$ design space, the design policy performs a finite number of steps to develop a VSR which is initiated by a single voxel. **(C)** A VSR is composed of 5 kinds of voxels.

In view of these challenges, we propose a Curriculum-based Co-design (CuCo) method for learning to design and control VSRs from easy to difficult (Figure 1). CuCo draws inspiration from Curriculum Learning (CL) that starting with easier tasks can help the agent learn better when presented with more difficult tasks later on (Bengio et al., 2009). The key to our approach is expanding the design space from a small size to the target size gradually through a predefined curriculum. Precisely, at each stage of the curriculum, we learn practical design and control patterns via Reinforcement Learning (RL) (Schulman et al., 2017), which is enabled by incorporating the design process into the environment and using differentiable policy representations. The converged morphology and the learned policies from last stage are inherited and then serve as the starting point for the next stage. Due to the exploitation of the previous knowledge of design and control, CuCo can quickly produce robust morphologies with better performance in larger dimensional settings.

To handle incompatible state-action spaces and make the control patterns transferable across various morphologies, we model the local observations of all voxels as a sequence and adopt the self-attention mechanism (Vaswani et al., 2017) in our control policy to capture the internal dependencies between voxels, which caters to the need for dynamically accommodating changes in morphologies. We also add material information to each voxel's observation, thus the control policy is conditioned on the robot's morphology, which eliminates the need for a population of prototypes and enables experience to be shared across different robots, bringing better sample efficiency and flexibility. For the design policy, we propose to utilize a Neural Cellular Automata (NCA) (Mordvintsev et al., 2020), which takes multiple actions to grow a robot from an initial seed (morphology). NCA encodes complex patterns in a neural network and generates different developmental outcomes while using a smaller set of trainable parameters.

In this paper, we make the following contributions: (1) We propose CuCo, a curriculum-based co-design approach for learning to design and control VSRs. (2) We demonstrate that the practical design and control patterns learned in small design spaces can be exploited to produce high-performing morphologies in larger dimensional settings. (3) We showcase that our NCA-based design policy and self-attention-based control policy can share design and control patterns across different sizes of morphologies, bringing better sample efficiency and flexibility. (4) Comprehensive experimental studies show that CuCo is more efficient in creating larger robots with better performance, in comparison to prior approaches that learn from scratch in the target design space.

## 2 RELATED WORK

**Brain-body Co-design.** Co-designing a robot to solve a given task has been challenging for robot designers for decades (Sims, 1994; Wang et al., 2019b; Nygaard et al., 2020; Medvet et al., 2021; Hejna et al., 2021; Ma et al., 2021). Related works can be roughly divided into two categories, gradient-free and gradient-based. Most gradient-free methods rely on artificial evolution to explore

the vast design space. Cheney et al. (2013) evolve VSRs with 3D voxels over materials and actuators. Bhatia et al. (2021) create a large-scale benchmark and several population-based co-design algorithms. Walker & Hauser (2021) evolve a sculpting adaptation system that allows VSRs to change their bodies to adapt to the environment. Talamini et al. (2021) propose a task-agnostic approach for automatically designing morphologies. While the population-based approaches are not easily trapped in local optima, they are sample inefficient and computationally expensive. Besides, most of these works rely on open-loop controllers, which simplify the control as a periodic sequence of actuation that depends only on the time (e.g., sinusoidal control signals), preventing robots from learning non-periodic behaviour. In contrast, gradient-based methods, especially reinforcement learning methods, have been conducted for optimizing rigid-bodied robots. Ha (2019) uses a population-based policy gradient method to improve robot design. Pathak et al. (2019) construct a self-organized modular robotic system that can generalize to unseen morphologies and tasks. Luck et al. (2019) train a surrogate for efficiently evaluating morphologies. Schaff et al. (2019) maintain a distribution over morphological parameters. Chen et al. (2020) learn a hardware-conditioned policy. Yuan et al. (2022) optimize a transform-and-control policy based on Graph Neural Network (GNN) in a fixed design space. Our work falls into the second category, instead, we focus on a scalable modular robot design space assigned by a curriculum. Within each learning stage, we unify the processes of design and control as a single decision-making process like (Yuan et al., 2022). However, both the design policy and the control policy of our work are exploited across stages as they are not restricted by robots' morphological topologies.

**Curriculum Learning.** Curriculum learning follows the spirit that training the agent with a series of tasks organized according to different complexity can speed up the convergence of the learning process (Bengio et al., 2009). It has achieved tremendous success in both RL and robotic communities (Li et al., 2020; Iscen et al., 2021; Jiang et al., 2021b). In RL, one of the keys to curriculum learning is how to generate and assign tasks. Dennis et al. (2020); Jiang et al. (2021a); Parker-Holder et al. (2022) focus on Unsupervised Environment Design (UED), which automatically constructs a distribution over environment variations, while Matiisen et al. (2020); Portelas et al. (2020) investigate teacher algorithms that monitor the learning progress and generate curricula to guide student algorithms to learn in difficult environments. In robotics, Wang et al. (2019a; 2020) generate an endless progression of increasingly challenging environments to evolve robust robots. Fang et al. (2021) propose a method to learn skills via automated generation of diverse tasks. In this work, we adopt a fixed curriculum by increasing the size of the robot design space according to the learning progress, as the number of possible robot morphologies increases exponentially with the expansion of each dimension of the design space.

**Learning a general Controller.** Another related line is learning generalizable control policies. Many such approaches utilize GNNs (Sanchez-Gonzalez et al., 2018; Wang et al., 2019b; Huang et al., 2020; Yuan et al., 2022) that condition the policy input on a graph representation related to the robot morphology. However, for modular robot systems, GNNs have the limitation of aggregating multi-hop information (Pigozzi et al., 2022), which makes modules not completely interchangeable, and the system could be influenced by information loss, greatly limiting their flexibility. Instead, Transformer (Vaswani et al., 2017) is not restricted by the input graph topology, as it can be characterized as a fully connected GNN with attentional aggregation (Kurin et al., 2021). It is shown that generalization can be further enhanced by modularization, due to the success of modeling dynamic structures via self-attention (Gupta et al., 2022; Trabucco et al., 2022).

## 3 BACKGROUND

**Voxel-based Soft Robots.** We investigate VSRs composed of multi-material cubic blocks (e.g., soft voxels, rigid voxels and actuator voxels) organized in a 2-D grid (Figure 1 (C)). The connectivity describes the connection pairs of adjacent voxels. During the simulation, voxels (except empty voxels) only sense locally, and based on the input sensory information, a controller outputs control signals to vary the volume of actuator voxels. Similar to biological tissues, they can be easily expanded and contracted to complete a wide variety of tasks (Bhatia et al., 2021).

**Reinforcement Learning.** In contrast to prior co-design works which seldom involve the learning of control, we use reinforcement learning to simultaneously train the design policy and the control policy. In the domain of RL, the problem is usually modeled as a Markov Decision Process (MDP),

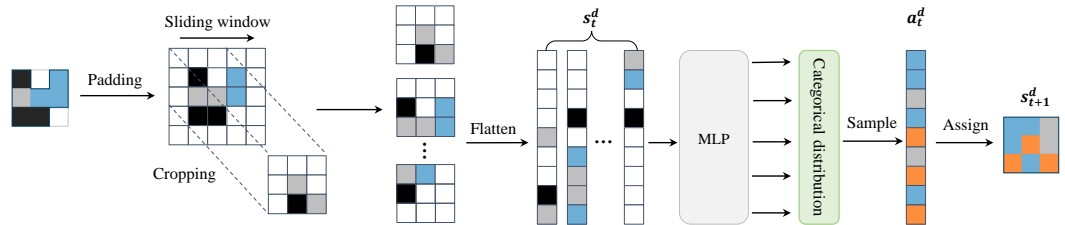

Figure 2: Visualization of a single design step. A multilayer perceptron is incorporated in NCA, which determines a cell's state by taking its current state and its neighboring cells' states as the input and mapping it to a categorical action distribution. As a result, each cell of the body is updated to the state with the sampled action.

which can be specified by a 5-tuple $(\mathcal{S}, \mathcal{A}, P, r, \gamma)$. With the state space $\mathcal{S}$ and the action space $\mathcal{A}$, $P : \mathcal{S} \times \mathcal{S} \times \mathcal{A} \to [0, 1]$ is the transition function of the environment, $r(s, a) : \mathcal{S} \times \mathcal{A} \to \mathbb{R}$ is the reward function, and the discount factor $\gamma \in (0, 1]$ specifies the degree to which rewards are discounted over time. The goal of the RL agent is to find a policy to maximize the total expected reward $\mathbb{E}_\pi [\sum_{t=0}^{\infty} \gamma^t r(s_t, a_t)]$, here the policy is represented by a deep neural network parameterized as $\pi_\theta(a_t|s_t)$, which maps from states to distributions over actions. In this work, we employ Proximal Policy Optimization (PPO) (Schulman et al., 2017), a popular RL algorithm base on Policy Gradient (PG). PPO uses Kullback-Leibler (KL) divergence to penalize the change between the old and the current policy to avoid instabilities from drastic changes in the policy's behaviour.

**Neural Cellular Automata.** Since we incorporate the design process into the environment, our objective is to find a design policy that can take in arbitrary VSRs' morphologies and output a series of actions to modify them. Typically, VSRs can be characterized as multi-cellular systems which develop from a small set of initial cells (Horibe et al., 2021). We represent our design policy as a Neural Cellular Automata (NCA) (Mordvintsev et al., 2020; Sudhakaran et al., 2022; Palm et al., 2022), which begins with some initial seeds and updates their states according to local rules parameterized by a multilayer perceptron, thus, NCA is agnostic to the robot's morphological topology, making it possible for scaling the learned rules to different design spaces.

**Self-attention.** Self-attention was proposed due to the challenges in handling long sequences (Vaswani et al., 2017). The goal is to dynamically track the relationships between components, which can be viewed as an adaptive weights mechanism. Given the input vectors $X$, $W^Q$, $W^K$ and $W^V$ are three trainable weight matrices. $Q, K$ and $V$ are the query, key, and value vectors where $Q = XW^Q, K = XW^K$ and $V = XW^V$. The importance scores are firstly computed by the dot product between $Q$ and $K$ and then scaled by a non-linear function. Self-attention is typically integrated into the Transformer block, which also contains a feed-forward neural network, and each of these two submodules has a residual connection around it and is succeeded by a normalization layer. Our work here specifically uses a Transformer encoder, enabling our control policy to dynamically accommodate changes in morphologies.

## 4  CuCo: Learning to Design and Control from Easy to Difficult

Given a modular design space (e.g., 7×7), exploring all possible combinations is unfeasible. The key to our approach is to expand this design space from a small size to the target size gradually through a predefined curriculum, which divides the whole optimization process into several stages (Figure 1 (A)). We maintain a design policy $\pi_{\theta_D}$ and a control policy $\pi_{\theta_C}$ from the beginning to the end, thus the overall policy is represented as $\pi_\theta = \{\pi_{\theta_D}(a_t^d|s_t^d, \Psi_t), \pi_{\theta_C}(a_t^c|s_t^c, \Psi_t)\}$ where $\Psi_t \in \{0, 1\}$ is a binary value indicating whether the robot is being created or controlled, $s_t^d$ and $s_t^c$ represent the state information received at each time step of design and control, respectively. Within each stage, we simultaneously optimize the design and control through reinforcement learning, which is enabled by unifying the two processes into a single MDP. Specifically, at the beginning of each episode, the design policy $\pi_{\theta_D}(a_t^d|s_t^d, \Psi_t)$ performs $N_D$ steps to develop a robot from an initial seed, and no rewards are assigned to these steps (Figure 1 (B)). The resulting robot is then presented

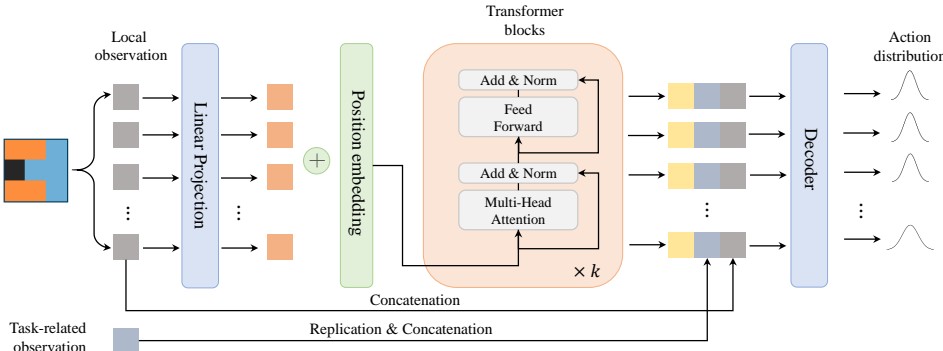

Figure 3: Visualization of the network architecture of the control policy. We model the local observations of all voxels as a sequence and utilize a Transformer encoder to process the current state.

in the simulator and controlled by $\pi_{\theta_C}(a_t^c|s_t^c, \Psi_t)$. Thus, the design policy will be informed by the future environmental rewards collected by the control policy under the designed morphology. After collecting the desired number of trajectories, the two policies are jointly updated using PPO (Schulman et al., 2017). From one stage to the next, the design space is expanded according to the curriculum and the converged morphology from last stage will serve as the new initial seed. The learned design policy and the control policy are inherited and then fine-tuned to produce larger bodies. In the remaining section, we describe details about each component of CuCo.

## 4.1 DESIGN POLICY

Figure 2 demonstrates a single design step when developing a VSR. The design space is surrounded by empty voxels, and each voxel is represented as a discrete value that corresponds to its material property (e.g., empty voxel=0, soft voxel=1, rigid voxel=2, horizontal actuator=3 and vertical actuator=4). The state vector $s^{d_i}$ for each voxel is composed of its type and the types of its neighborhood (e.g., Neumann or Moore neighborhood) (Horibe et al., 2021). Practically, let $N$ be the size of the design space, the MDP state of a design step is represented by $s_t^d = \{s_t^{d_1}, s_t^{d_2}, ..., s_t^{d_N}\}$. Here, the design policy $\pi_{\theta_D}(a_t^d|s_t^d, \Psi_t)$ is a multilayer perceptron parameterized by $\theta_D$, which maps $s_t^d$ to a categorical action distribution created by the output logits. The dimension of the input layer corresponds to the number of a cell's neighbors (e.g, 9 in Figure 2), and the dimension of the output layer corresponds to the total number of material types (e.g, 5 in Figure 2). During training, the new cell state is determined by sampling from the distribution. While evaluating, the design policy selects the action that corresponds to the highest logit value.

At the beginning of each episode of the first learning stage, a single voxel placed at the center of the design space is used as the initial seed, and the rest voxels' states are set to zero. After $N_D$ design steps, all voxels are updated according to the policy's outputs. In the next learning stage, the converged morphology from last stage will be placed at the center of the design space as the new initial seed. Meanwhile, the weights of the learned design policy are also inherited. Since the morphology defines the connectivity, here, we let the design policy learn the connectivity on its own. We do not assign any design-related rewards to the design steps and terminate the episode if the resulting design does not meet the connectivity or has no actuators.

## 4.2 CONTROL POLICY

Once a robot is designed, it is presented in the simulator and consumed by the control policy $\pi_{\theta_C}(a_t^c|s_t^c, \Psi_t)$. We model the local observations of all voxels as a sequence. The input state at time step $t$ is represented as $s_t^c = \{s_t^v, s_t^g\}$, where $s_t^v = \{s_t^{v_1}, s_t^{v_2}, ..., s_t^{v_N}\}$, $s_t^{v_i}$ is composed of each voxel's local information which contains the relative position of its four corners with respect to the center of mass of the robot and its material information. $s_t^g$ is the task-related observation such as terrain information of the environment and goal-relevant information.

To capture the internal dependencies between voxels and dynamically accommodate changes in morphologies, we incorporate the self-attention mechanism into the control policy. As shown in Figure 3, we apply a learnable position embedding layer after the linear projection layer. The local observation, the output feature of the Transformer and the task-related observation are concatenated before passing them through a decoder. In our work, the control policy outputs the mean $\mu$ of the continuous action, and we use a constant standard deviation (a fixed diagonal covariance matrix $\Sigma$) to create the Gaussian distribution $\pi_{\theta_C}(a_t^c|s_t^c, \Psi_t) = \mathcal{N}(\mu(s_t^c, \Psi_t), \Sigma)$. All actions are clipped within the range of $[0.6, 1.6]$. During training, the action is randomly sampled from the distribution. While evaluating the policy, only the mean action is used for control.

### 4.3 POLICY UPDATE AND THE CURRICULUM

We simultaneously train the design policy and the control policy using PPO, which is based on the Actor-Critic architecture. In our work, the critic network has the same architecture as the control policy network. It estimates the value function $\mathbb{E}_{\pi_\theta}[\sum_{t=0}^{T} \gamma^t r]$ which suggests a possible policy distribution. Here, we estimate the value for the whole morphology by averaging the value per voxel. With the policy gradient method, all policies are updated in the direction of maximizing the predicted value. When estimating the value of a design step, we set the environment state $s^c$ to zero. We apply a predefined curriculum according to the training progress as the number of possible robot designs increases exponentially with each additional cube. During training, we increase the dimension of the design space when the target number of policy iterations is reached.

## 5 EMPIRICAL EVALUATION

In this section, we demonstrate the effectiveness of CuCo on various tasks. We seek to answer the following questions: (1) Does CuCo provide an effective mechanism for learning to design and control VSRs? (2) How does CuCo compare to prior approaches which learn from scratch in the space of target size in terms of the final performance and the produced morphologies? (3) What impact does each component of CuCo bring to the learning process?

### 5.1 EXPERIMENT SETTINGS

We create a modular robot design space described in Section 4 and 8 task-related environments based on Evolution Gym (Bhatia et al., 2021). The environment details can be found in Appendix A and our codes are available online[1]. We compare CuCo with a number of approaches, both RL-based and population-based. All baselines are introduced as follows:

**CuCo-NCU**: CuCo-NCU utilizes the same network architectures as CuCo but eliminates the curriculum component. It learns to design and control from scratch in the space of target size. This method helps us to investigate the effectiveness of the curriculum.

Additionally, we compare CuCo with three evolutionary methods proposed in (Bhatia et al., 2021). The inner loop of control optimization is also driven by PPO, and the outer loop of morphology optimization is implemented by three state-of-the-art design optimization algorithms.

**GA:** Genetic Algorithm directly encodes the VSR's morphology as a vector where each element is tailored to the voxel's material property in order. It keeps a proportion of robots with high fitness as survivors at the end of each generation, and then the survivors undergo mutation conducted by randomly changing each voxel.

**CPPN-NEAT:** In this method, the morphology of VSR is indirectly encoded by a Compositional Pattern Producing Network (CPPN) (Stanley, 2007; Cheney et al., 2013), which relies on having access to the whole design space. The training of CPPN is accomplished by NeuroEvolution of Augmenting Topologies (NEAT) (Stanley & Miikkulainen, 2002).

**BO:** Bayesian Optimization (BO) is widely used to help solve black-box problems. It trains a surrogate model based on Gaussian Process (GP) to alleviate the computational burden of expensive fitness evaluation (Kandasamy et al., 2018).

---

[1] https://github.com/Yuxing-Wang-THU/ModularEvoGym

Table 1: Mean and standard error of all results across environments and baselines. Results with the highest mean are shown in bold. ∗ indicates that the method reaches the upper bound of the score.

| Environment | CuCo | CuCo-NCU | GA | CPPN-NEAT | BO |
|---|---|---|---|---|---|
| Walker | ∗10.46 ± 0.01 | 8.90 ± 2.01 | ∗10.45 ± 0.02 | ∗10.46 ± 0.01 | 9.38 ± 1.13 |
| Pusher | **11.74 ± 1.43** | 1.73 ± 2.19 | 7.88 ± 1.30 | 9.66 ± 1.24 | 8.73 ± 1.16 |
| Jumper | **13.08 ± 1.09** | 0.79 ± 1.98 | 1.85 ± 1.31 | 3.82 ± 2.33 | 1.28 ± 0.78 |
| UpStepper | **4.40 ± 0.28** | 2.45 ± 1.70 | 2.58 ± 0.74 | 3.65 ± 0.50 | 2.43 ± 0.53 |
| PlatformJumper | **3.13 ± 0.93** | 1.58 ± 0.19 | 2.03 ± 0.50 | 2.36 ± 0.07 | 1.60 ± 0.03 |
| Thrower | **3.59 ± 1.61** | 2.88 ± 0.60 | 1.75 ± 0.19 | 1.60 ± 0.28 | 1.91 ± 0.21 |
| Climber | 0.54 ± 0.09 | 0.28 ± 0.28 | 0.58 ± 0.66 | **1.54 ± 0.52** | 0.83 ± 0.91 |
| Lifter | 0.00 ± 0.00 | 0.00 ± 0.00 | 0.04 ± 0.04 | 0.00 ± 0.00 | **0.10 ± 0.16** |

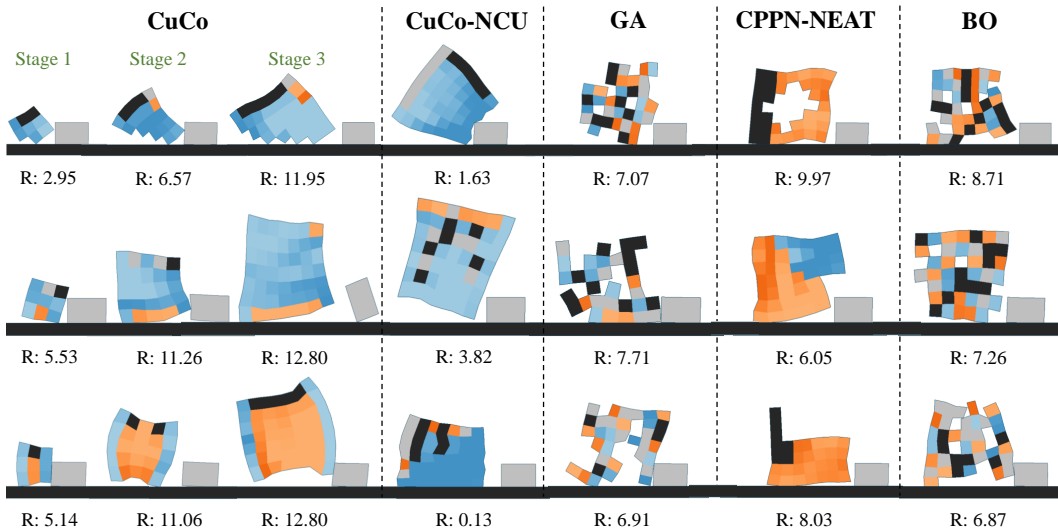

Figure 4: Visualization of converged morphologies of all methods across three different runs in Pusher. Below each VSR is its average performance over 5 tests. More visual results can be found in this video.

Following the convention in the literature (Cheney et al., 2013; Horibe et al., 2021), we consider a 2-D design space where all dimensions are equal in size. The curriculum defines three stages $(3 \times 3 \rightarrow 5 \times 5 \rightarrow 7 \times 7)$ for Walker, Pusher, Jumper and Lifter. For the remaining tasks, the curriculum is set to $(3 \times 3 \rightarrow 5 \times 5)$. We run all experiments with the same number of time steps. For CuCo, this amounts to 1000 policy updates per stage. We employ a population of 12 agents for all population-based methods and report the performance of the best individual. Further implementation details can be found in Appendix D.

## 5.2 OVERALL PERFORMANCE

In Table 1, we list the final performance of each method, and the result is reported over 7 different runs per environment. Learning curves for all methods and visualization results of converged morphologies can be found in Appendix B.1. We observe that within error margins, CuCo outperforms the baselines in 5 of the 8 environments, while only CPPN-NEAT and BO can achieve limited success in Climber and Lifter, respectively. In Section 6, we discuss in detail these two cases. Without the guidance of the curriculum, CuCo-NCU performs poorly on most tasks since learning to design and control from scratch in a high-dimensional space is challenging.

We demonstrate the converged morphologies of all methods across three different runs and their corresponding performance in Figure 4, and for CuCo, we visualize the converged design of each stage. Direct encoding methods like GA and BO prefer horse-like morphologies composed of ran-

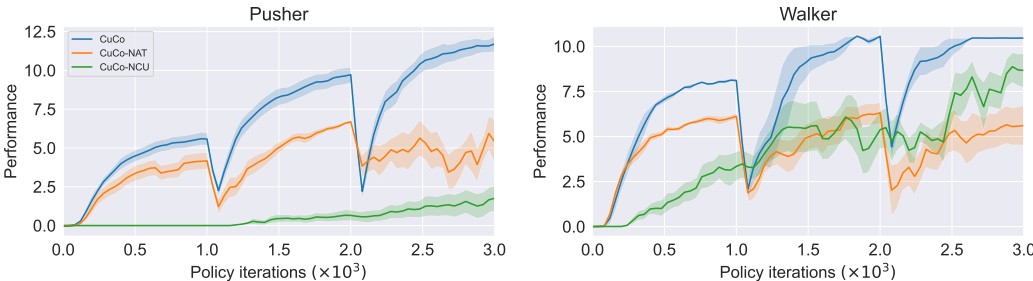

Figure 5: Ablations of the self-attention mechanism and the curriculum.

domly organized voxels. However, voxels from the last two robots created by GA often collide with each other, preventing robots from learning efficient control policies. CuCo-NCU fails to discover practical policies. In contrast, we find that CuCo can develop beneficial homogeneous tissue structures by module reuse, which can also be observed in many creatures, such as insects and vertebrates with several repeated body segments determined by homologous genes in different species (Dellaert, 1995). When the robot is moving, we can see that groups of voxels are expanded or compressed simultaneously, leading to coordinated behaviour.

Interestingly, VSRs designed by CuCo obtain different skills for pushing a box. The first rhino-like robot chooses to push the box with its "horn". The second robot makes full use of vertical actuators to let the box roll forward, while the third one uses horizontal actuators to pop the box out. As shown in the video, CuCo also produces robots that can keep the balance as well as find a better launch angle when throwing the box. Although GA and BO produce VSRs with arms, they are not strong enough to throw the box further. CPPN-NEAT generates much simpler patches based on the global information of the design space. In Appendix B.2, we further investigate differences between CPPN and NCA when chosen as the indirectly encoded design policy for CuCo. In sum, CuCo holds great potential for exploiting practical design and control patterns learned in small spaces to produce high-performing robots in larger dimensional settings.

## 5.3 ABLATIONS

We perform ablation studies to take a closer look at the behaviour of CuCo. Firstly, we demonstrate the learning curves of CuCo and CuCo-NCU (Figure 5) to show that following an easy-to-difficult curriculum is much more efficient than learning from scratch in the target design space. Especially in Pusher, CuCo-NCU spends a large amount of time learning the connectivity. Secondly, we remove the self-attention mechanism used in our control policy and construct a new ablation **CuCo-NAT**. Evidently, after removing the self-attention mechanism, the control policy fails to dynamically adapt to morphological changes, resulting in much lower performance. In Appendix C, we analyze the attention matrices that CuCo learns. These two ablations indicate that the curriculum and the self-attention mechanism are indeed crucial for finding better solutions.

Another key feature of CuCo is the mechanism of inheritance. Between the two learning stages, the converged morphology, the learned design and control policies are inherited. We perform additional ablation studies to measure the impact of the inheritance on the subsequent learning process. We design the following ablations: (1) **CuCo-NIM**: No inheritance of the converged morphology from last stage; (2) **CuCo-NID**: No inheritance of the design policy. We reset the parameters of the design policy at the end of each stage; (3) **CuCo-NIC**: No inheritance of the control policy. We reset the parameters of the control policy at the end of each stage; (4) **CuCo-NIDC**: No inheritance of the design policy and the control policy. We reset all network parameters at the end of each stage. The learning curves of all ablations are shown in Figure 6, and for better visualization, we plot these curves from the second learning stage. By comparing CuCo with CuCo-NID and CuCo-NIDC, it is worth noting that the inheritance of the design policy plays a critical role during the learning process. Even if the control policy is reset (CuCo-NIC), the performance is less affected. Additionally, the inheritance of the converged morphology also contributes to the learning process, as new morphologies can be quickly built by expanding the old morphology.

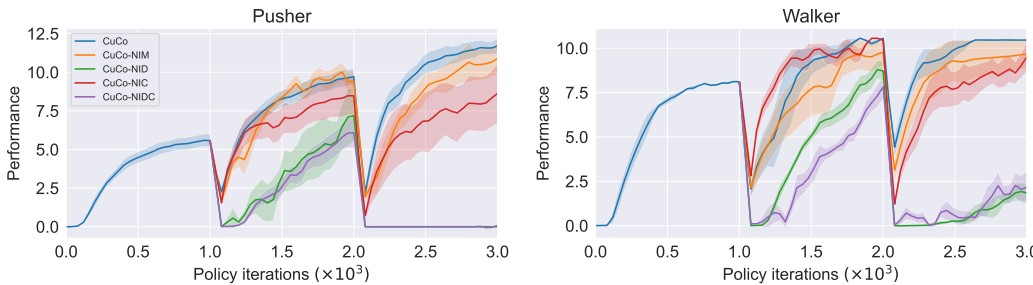

Figure 6: Ablations of the inheritance mechanism. The plots show that the inheritance of the design policy plays a critical role during the learning process.

## 6 DISCUSSION AND CONCLUSION

In this work, we proposed CuCo, a curriculum-based co-design method for learning to design and control voxel-based soft robots through an easy-to-difficult process. Compared with previous approaches that learn from scratch in the target design space, our method is more efficient in producing larger robots with better performance as it follows a predefined curriculum and effectively exploits the knowledge of design and control obtained in small dimensional settings. Empirical evaluations on various robot control tasks show that CuCo outperforms prior methods in terms of the final performance, producing intriguing robot designs with repeated body segments.

However, compared with GA and BO, we also observe that CuCo tends to produce more regular shapes, resulting in failures on some complex tasks (e.g., Climber and Lifter) which may require robots to have irregular body structures (e.g., thin limbs or fingers). We speculate that there are two reasons for the failure: (1) **Design**. In our work, NCA is encouraged to learn the connectivity on its own and we do not assign any design-related rewards. Specifically, we check the connectivity at each design step and terminate the episode if the resulting design does not meet the connectivity or has no actuator voxels. Therefore, NCA becomes conservative in using empty voxels. Due to the mechanism of inheritance, this kind of design pattern would be further extended in larger design spaces, restricting the exploration of irregular morphologies. (2) **Control**. For some challenging tasks, when it is hard for the control policy to explore and exploit the environmental reward, the design policy is also hard to be informed because we formulate the processes of design and control into a single MDP. Fortunately, the failures also suggest that we can add design rewards to guide NCA to explore some counterintuitive structures, and using more complex NCA architectures (Sudhakaran et al., 2022) may also help the exploration of irregular designs. Hence an important future work is creating an efficient mechanism for balancing the exploration and exploitation abilities of NCA.

During the learning process, we do not constrain the number of modules for each type of material. Thus, the optimization is uncontrolled. A future direction would be to investigate the constrained co-design method. We found that CuCo performs well under the fixed curriculum, a promising direction of future work would involve extending our method to automatic curriculum learning. In CuCo, two kinds of developmental processes happen on different timescales. Within each stage, the robots grow under the restriction of body size. From one stage to the next, robots will become bigger and stronger. However, size is not the only morphological characteristic of living creatures. An important line of exploration is to extend CuCo to help study the morphogenesis of virtual creatures. Co-designing physical VSRs (Kriegman et al., 2019; 2020) using our method is expected to further extend the horizon.

## 7 ACKNOWLEDGEMENT

We thank the anonymous reviewers for their helpful comments in revising the paper. This work is partially supported by Tencent Rhino-Bird Research Elite Program (2022).

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

## A    ENVIRONMENT DETAILS

In this section, we provide details about the tasks simulated in Evolution Gym platform (Banarse et al., 2019).

**Position.** $p^o$ is a 2-dim vector that represents the position of the center of mass of an object $o$ in the simulation at time $t$. $p_x^o$ and $p_y^o$ are $x$ and $y$ components of this vector, respectively. $p^o$ is calculated by averaging the positions of all the point-masses that make up object $o$ at time $t$.

**Velocity.** $v^o$ is a 2-dim vector that represents the velocity of the center of mass of an object $o$ in the simulation at time $t$. $v_x^o$ and $v_y^o$ are $x$ and $y$ components of this vector, respectively. $v^o$ is calculated by averaging the velocities of all the point-masses that make up object $o$ at time $t$.

**Orientation.** $\theta^o$ is a 1-dim vector that represents the orientation of an object $o$ in the simulation at time $t$. Let $p_i$ be the position of point mass $i$ of object $o$. $\theta^o$ is computed by averaging over all $i$ the angle between the vector $p_i - p^o$ at time $t$ and time 0. This average is a weighted average weighted by $||p_i - p^o||$ at time 0.

**Other observations.** $h_b^o(d)$ is a vector of length $(2d+1)$ that describes elevation information around the robot below its center of mass. More specifically, for some integer $x \leq d$, the corresponding entry in vector $h_b^o(d)$ will be the highest point of the terrain which is less than $p_y^o$ between a range of $[x, x+1]$ voxels from $p_x^o$ in the $x$-direction.

### A.1    WALKER

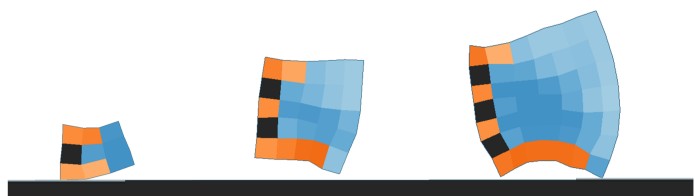

Figure 7: Walker environment.

In this task, the robot is rewarded by walking as far as possible on flat terrain. The target design space is $7 \times 7$. The task-specific observation is $v^{robot}$, and the reward $R$ is:

$$R = \Delta p_x^{robot} \tag{1}$$

which rewards the robot for moving in the positive $x$-direction. The robot receives a reward of 1 for reaching the end of the terrain. The episode duration reaches a 500 time steps.

### A.2    PUSHER

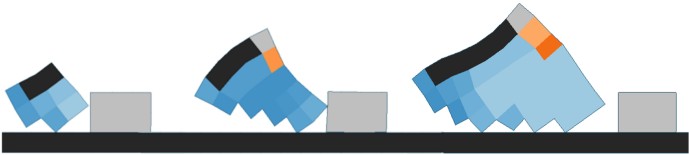

Figure 8: Pusher environment.

In this task, the robot pushes a box initialized in front of it. The target design space is $7 \times 7$. The task-specific observation is formed by concatenating vectors $\{v^{box}, p^{box} - p^{robot}, v^{robot}\}$, and the reward $R = R_1 + R_2$, where $R_1$ is:

$$R_1 = 0.5 \cdot \Delta p_x^{robot} + 0.75 \cdot \Delta p_x^{box} \tag{2}$$

which rewards the robot and box for moving in the positive $x$-direction, and $R_2$ is:

$$R_2 = -\Delta |p_x^{box} - p_x^{robot}| \tag{3}$$

which penalizes the robot and box for separating in the $x$-direction. The robot also receives a one-time reward of 1 for reaching the end of the terrain. The episode duration reaches a 500 time steps.

### A.3 UPSTEPPER

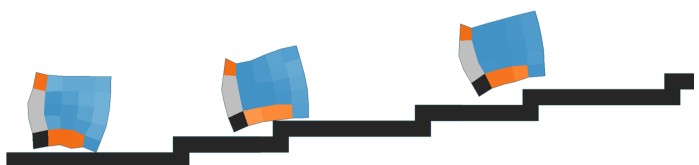

Figure 9: UpStepper environment.

In this task, the robot climbs up stairs of varying lengths. The target design space is $5 \times 5$. The task-specific observation is formed by concatenating vectors $\{v^{robot}, \theta^{robot}, h_b^{robot}(5)\}$, and the reward $R$ is:

$$R = \Delta p_x^{robot} \tag{4}$$

which rewards the robot for moving in the positive $x$-direction. The robot also receives a one-time reward of 2 for reaching the end of the terrain. The episode duration reaches a 600 time steps.

### A.4 JUMPER

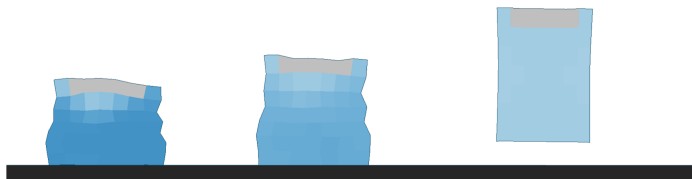

Figure 10: Jumper environment.

In this task, the robot jumps as high as possible in place on flat terrain. The target design space is $7 \times 7$. The task-specific observation is formed by concatenating vectors $\{v^{robot}, h_b^{robot}(2)\}$, and the reward $R$ is:

$$R = 10 \cdot \Delta p_y^{robot} - 5 \cdot |\Delta p_x^{robot}| \tag{5}$$

which rewards the robot for moving in the positive $y$-direction and penalizes the robot for any motion in the $x$-direction. The episode duration reaches a 500 time steps.

### A.5 PLATFORMJUMPER

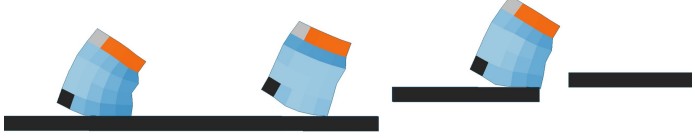

Figure 11: PlatformJumper environment.

In this task, the robot traverses a series of floating platforms at different heights. The target design space is $5 \times 5$. The task-specific observation is formed by concatenating vectors $\{v^{robot}, \theta^{robot}, h_b^{robot}(5)\}$, and the reward $R$ is:

$$R = \Delta p_x^{robot} \tag{6}$$

which rewards the robot for moving in the positive $x$-direction, The robot also receives a one-time penalty of $-3$ for rotating more than 90 degrees from its originally orientation in either direction or for falling off the platforms (after which the environment resets). The episode duration reaches a 1000 time steps.

## A.6 THROWER

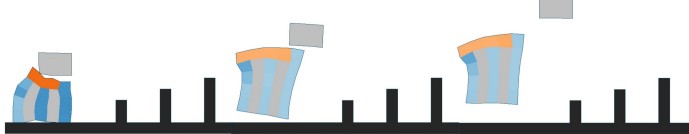

Figure 12: Thrower environment.

In this task, the robot throws a box initialized on top of it. The target design space is $5 \times 5$. The task-specific observation is formed by concatenating vectors $\{v^{box}, p^{box} - p^{robot}, v^{robot}\}$, and the reward $R = R_1 + R_2$, where $R_1$ is:

$$R_1 = \Delta p_x^{box} \qquad (7)$$

which rewards the box for moving in the positive $x$-direction, and $R_2$ is:

$$R_2 = \begin{cases} -\Delta p_x^{robot} & if \quad p_x^{robot} \geq 0 \\ \Delta p_x^{robot} & otherwise \end{cases} \qquad (8)$$

which penalizes the robot for moving too far from $x = 0$ when throwing the box. The episode duration reaches a 300 time steps.

## A.7 CLIMBER

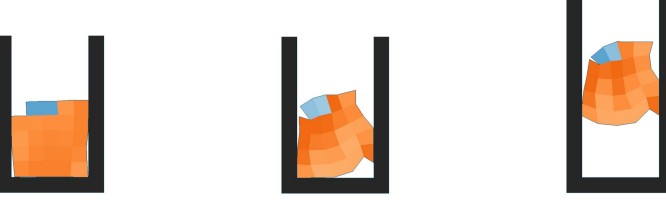

Figure 13: Climber environment.

In this task, the robot climbs as high as possible through a flat, vertical channel. The target design space is $5 \times 5$. The task-specific observation is $v^{robot}$, and the reward $R$ is:

$$R = \Delta p_y^{robot} \qquad (9)$$

which rewards the robot for moving in the positive $y$-direction. The robot also receives a one-time reward of 1 for reaching the end of the channel. The episode duration reaches a 400 time steps.

## A.8 LIFTER

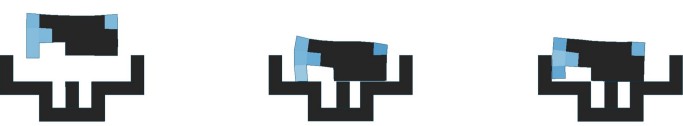

Figure 14: Lifter environment.

In this task, the robot lifts a box out of a hole. The target design space is $7 \times 7$. The task-specific observation is formed by concatenating vectors $\{p^{box} - p^{robot}, v^{robot}, v^{box}, \theta^{box}\}$, and the reward $R = R_1 + R_2 + R_3$, where $R_1$ is:

$$R_1 = 10 \cdot \Delta p_y^{box} \qquad (10)$$

which rewards the robot for moving the beam in the positive $y$-direction, and $R_2$ is:

$$R_2 = -10 \cdot \Delta |g_x - p_x^{box}| \qquad (11)$$

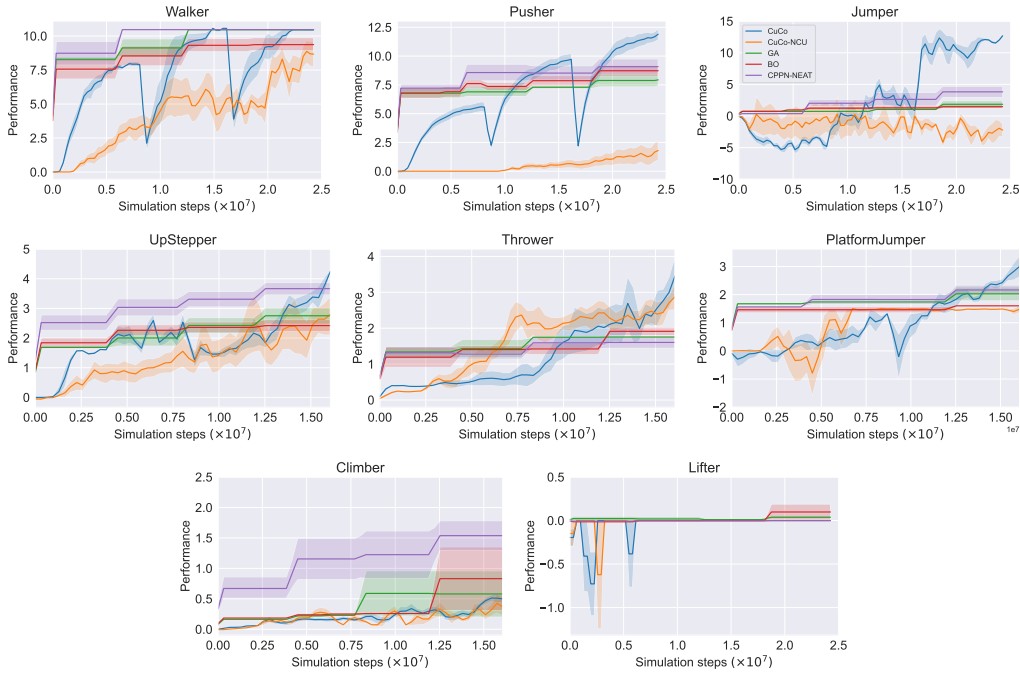

Figure 15: Learning curves of all methods.

which penalizes the robot for moving the box away from a goal $x$-position, $g_x$. This ensures that the robot lifts the box straight up, and $R_3$ is:

$$R_3 = \begin{cases} 0 & if \quad p_y^{robot} \geq t_y \\ 20 \cdot \Delta p_y^{robot} & otherwise \end{cases} \tag{12}$$

which penalizes the robot for falling below a threshold height $t_y$ (at which point the robot has fallen into the hole). The episode duration reaches a 300 time steps.

# B ADDITIONAL EXPERIMENTS

## B.1 LEARNING CURVES AND MORPHOLOGY RESULTS

Figure 15 plots the learning curves of all methods across various environments. In Figure 16, we provide visualization results of morphologies constructed using CuCo and baselines. In Figure 17, we demonstrate the development of robot morphologies during CuCo's learning process in UpStepper and Pusher environments, respectively.

## B.2 THE CHOICE OF DESIGN POLICY

We perform an additional experiment to investigate the importance of using the NCA architecture as CuCo's design policy. Here, we replace NCA with a Compositional Pattern Producing Network (CPPN) (Stanley, 2007) which takes the global information (e.g., $x$ and $y$ coordinates of the design space) as the input and directly generates the robot design parameters. We refer to this method as CuCo-CPPN. Note that the converged morphology and the design policy are not inheritable in CuCo-CPPN as the input dimension of CPPN is related to the design space. Figure 18 shows the learning curves of CuCo-NCA and CuCo-CPPN in Pusher and Walker. Evidently, CuCo-NCA outperforms CuCo-CPPN in terms of convergence speed and final performance. It shows that NCA can encode practical design patterns and successfully leverage them in larger design spaces, while CPPN does not significantly facilitate the learning process.

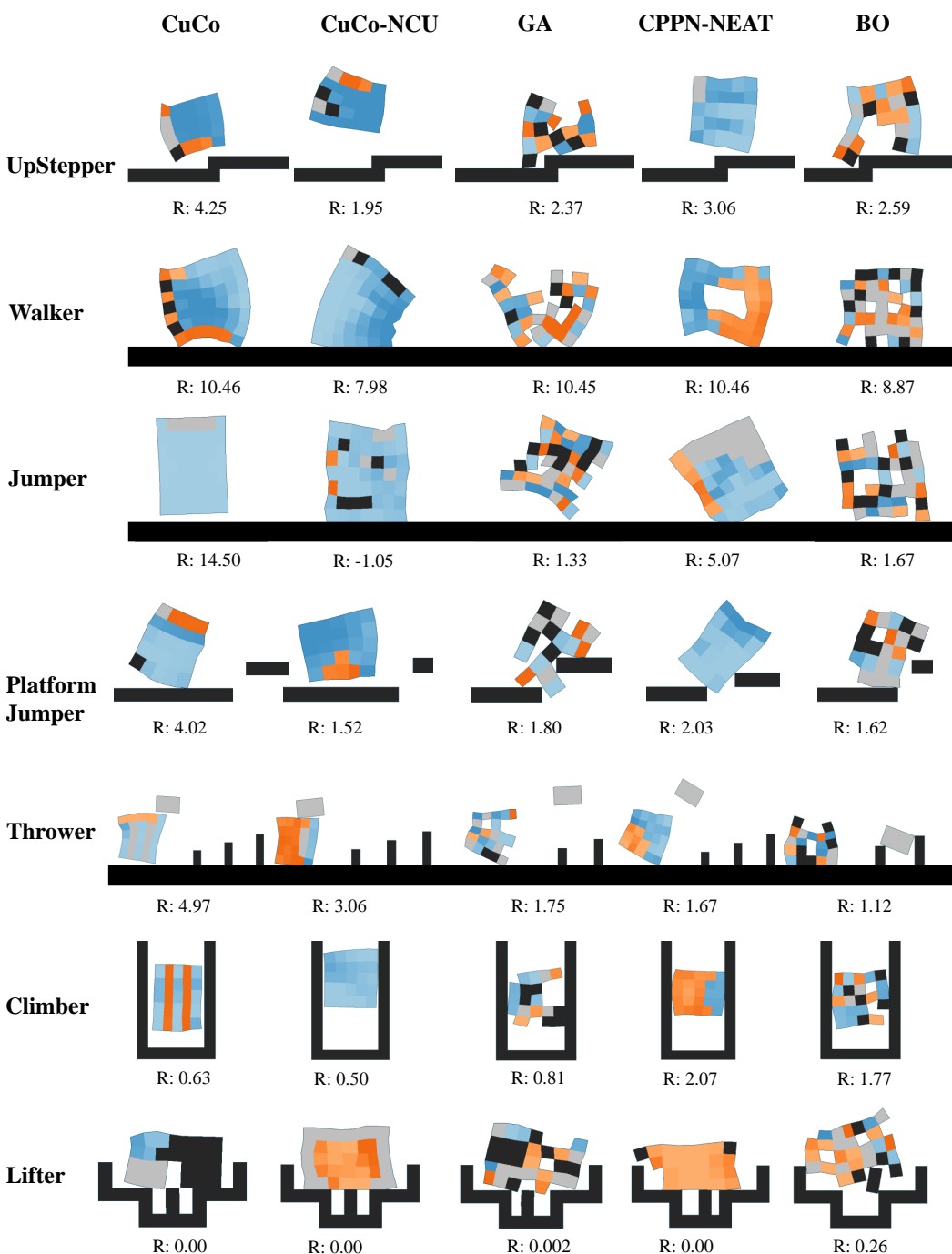

Figure 16: Visualization of converged morphologies of all methods across various environments. Below each VSR is its average performance over 5 tests.

## C ATTENTION MATRICES ANALYSIS

Self-attention brings better interpretability than multilayer perceptron. In this section, we analyze the attention matrices that CuCo learns. We use only one Transformer encoder layer, thus we visualize the generated attention matrix after one input state passes through the attention layer. Figure 19 demonstrates the learned attention matrices of the critic network when designing a robot, where the voxel 9 is assigned with more attention scores. Figure 20 shows attention matrices produced by

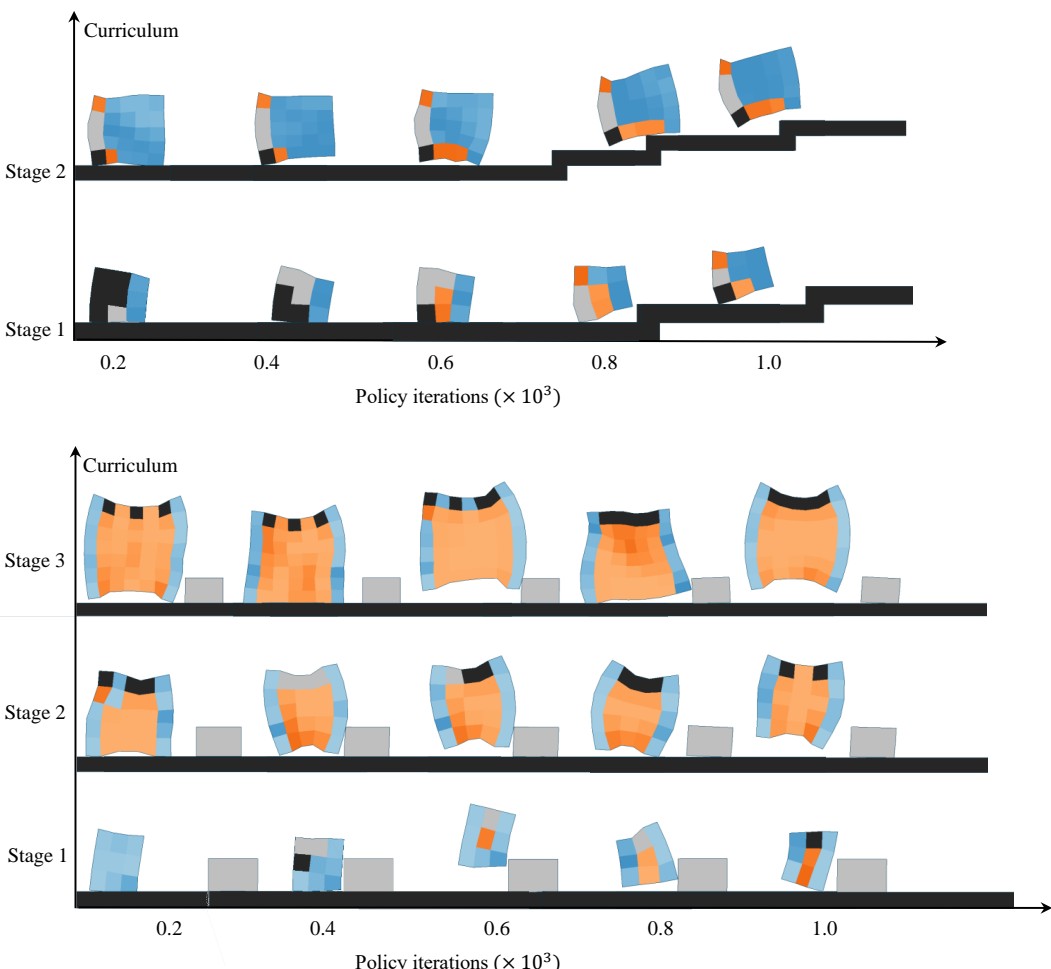

Figure 17: Visualization of CuCo's learning process. The $x$-axis represents the number of policy iterations and the $y$-axis represents the curriculum. We show morphologies produced within each stage from left to right.

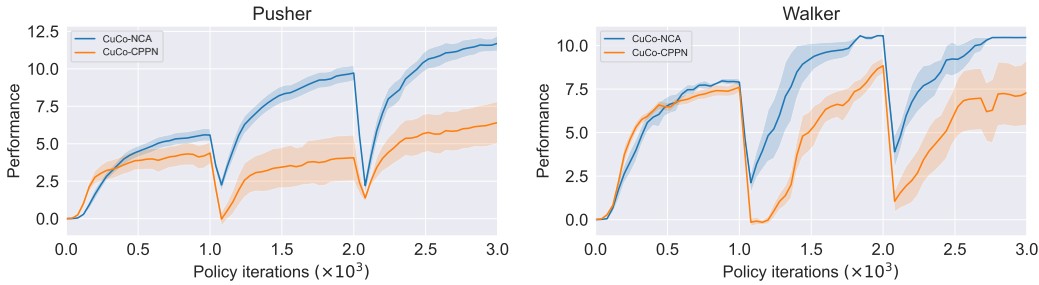

Figure 18: A comparison of using NCA and CPPN as the design policy. The results are averaged over 7 runs and suggest that the NCA architecture contributes to the performance greatly.

the control policy network. The color of each attention score tells the strength of the compatibility between inputs and interprets what is driving the current behaviour. When the robot's front foot (voxel 9) or the rear foot (voxel 7) touches the ground, the corresponding voxels are assigned with greater wights, which is consistent with humans' intuition and common sense.

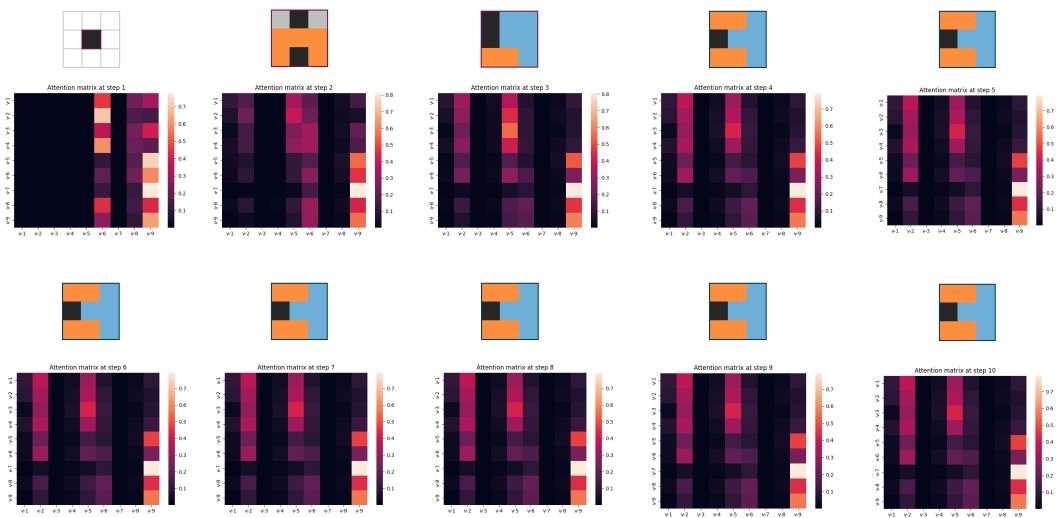

Figure 19: Attention matrices of 10 design steps.

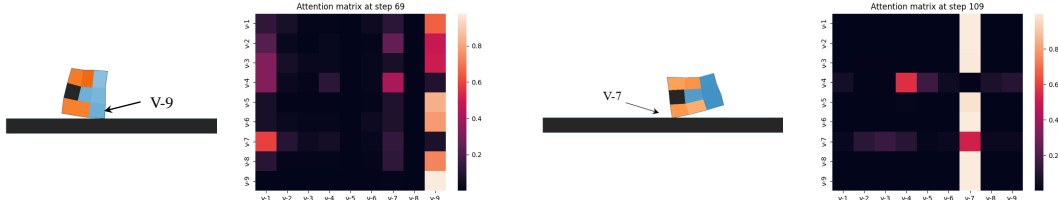

Figure 20: Attention matrices of 2 control steps.

# D  IMPLEMENTATION DETAILS

We use PyTorch (Paszke et al., 2019) to implement our proposed method. We take the official implementation of Transformer from Pytorch and add the position embedding. All hyperparameters for CuCo are listed in Table 2. For baseline algorithms, we use the official implementations of GA, CPPN-NEAT and BO from Evolution Gym (Bhatia et al., 2021). We employ a population of 12 agents for all population-based methods. Following the convention in the literature, we use multiple CPU threads to accelerate the training of PPO. For all the environments used in this paper, it takes around 1 day to train our model on a computer with 12 CPU cores and an NVIDIA RTX 3090 GPU.

Table 2: Hyperparameters of CuCo.

| | Hyperparameter | Value |
|---|---|---|
| | Use GAE | True |
| | GAE parameter $\lambda$ | 0.95 |
| | Learning rate | $2.5 \cdot 10^{-4}$ |
| | Use linear learning rate decay | True |
| | Clip parameter | 0.1 |
| | Value loss coefficient | 0.5 |
| | Entropy coefficient | 0.01 |
| | Time steps per rollout | 2048 |
| | Num processes | 4 |
| PPO | Optimizer | Adam |
| | Evaluation interval | 10 |
| | Discount factor $\gamma$ | 0.99 |
| | Clipped value function | True |
| | Observation normalization | True |
| | Observation clipping | $[-10, 10]$ |
| | Reward normalization | True |
| | Reward clipping | $[-10, 10]$ |
| | Policy epochs | 8 |
| | Policy iterations per stage | 1000 |
| | Neighborhood | Moore |
| | Input dimension | 9 |
| | Output dimension | 5 |
| NCA | Growth factor | False |
| | Hidden layers | $(64, 64)$ |
| | Activation function | Tanh |
| | Design steps | 10 |
| | Number of layers | 1 |
| | Number of attention heads | 1 |
| Transformer | Embedding dimension | 64 |
| | Feedforward dimension | 128 |
| | Non linearity function | ReLU |
| | Dropout | 0.0 |

