# OpenReview forum: "Curriculum-based Co-design of Morphology and Control of Voxel-based Soft Robots"
_ICLR.cc/2023/Conference — ICLR 2023 poster_

### Official Review · Reviewer_vM9V · 2022-10-23

**Confidence:** 5
**Correctness:** 4
**Technical Novelty And Significance:** 3
**Empirical Novelty And Significance:** 3
**Recommendation:** 6

**Clarity, Quality, Novelty And Reproducibility:**

The paper is clearly written and sufficiently novel, with reasonable technical approaches. For reproducibility, it's not clear to me - depending on if the authors are willing to release the source code.

**Strength And Weaknesses:**

Strengths:
- A novel and sound technical approach. Each component of the approach is intuitive and effective (curriculum, transformer-based generalizable control policy, NCA-based design policy).
- Good empirical performance with abundant ablation studies (Sec 5.4), which explains the necessity of all the algorithmic components.

Weaknesses:
- This paper only performs evaluations on 4 environments/tasks from Evolution Gym [Bhatia et al. 2021], which is not sufficient enough in my opinion to show the generalizability of the proposed approach. I would suggest at least evaluate on 6-10 environments from Evolution Gym, to convince people that the results are not cherry-picked.
- Limitations or failure cases are not discussed well in the paper, and I think this is crucial. Is there any case in which CuCo fails to beat the baselines?

Questions:
- By looking at the results, CuCo tends to produce regular shapes for the robot design compared to GA. It does not appear to evolve robots with some thin limb structures. However, my intuition tells me that for some complex tasks (such as climbing/throwing/lifting in environments), thin structures like limbs or fingers are the key to success. Could you please evaluate your method on those environments in Evolution Gym as well and report the performance?
- Could you show the performance/convergence curves for the baseline methods as well?

**Summary Of The Paper:**

This paper proposes a novel curriculum-based co-design method for evolving soft voxel-based robots. The main contributions are:
- A novel curriculum mechanism that effectively evolves robots from simple to large design spaces.
- A transformer-based control policy representation, enabling adaptation to arbitrary dimensional observations due to morphology changes.
- A novel design policy based on Neural Cellular Automata (NCA), making end2end training of both design and control policy possible.
- Impressive visual behavior and empirical performance compared to baseline methods with abundant ablation studies.

**Summary Of The Review:**

This paper proposes a novel method with sufficiently new insights into soft robot co-design. While the approach is theoretically sound, on the practical side, I still have several concerns (evaluations on more environments, discussions on limitations and failure cases).

---

> ### Author Response · Authors · 2022-11-18
> **Response to Reviewer vM9V (Part 1/2)**
>
> We are glad that the reviewer found our work impressive and appreciated the novelty and soundness of CuCo. Inspired by the reviewer, we added more experiments on different tasks (Section 5.2) and discussions of the limitations of our method (Section 6). Moreover, we are willing to release all the source code and sincerely hope that our work can contribute to the development of robot co-design. For the convenience of the reviewer, we reiterate the comments below.
>
> >**[Q1]** _This paper only performs evaluations on 4 environments/tasks from Evolution Gym [Bhatia et al. 2021], which is not sufficient enough in my opinion to show the generalizability of the proposed approach. I would suggest at least evaluate on 6-10 environments from Evolution Gym, to convince people that the results are not cherry-picked._
>
> **[A1]** Thanks for your suggestion. We have run additional experiments on 4 tasks (PlatformJumper, Thrower, Climber and Lifter) and showed the results in Table 1 of Section 5.2. Overall, CuCo outperforms all baselines on 6 of 8 tasks, and we believe that these results further strengthen the significance of our work. Inspired by the reviewer, we discussed 2 failure cases in Section 6. We encourage the reviewer to examine our video of supplementary results: https://files.catbox.moe/yubr3s.mp4
>
> >**[Q2]** _Limitations or failure cases are not discussed well in the paper, and I think this is crucial. Is there any case in which CuCo fails to beat the baselines?_
>
> **[A2]** Thanks for pointing this out. We have rephrased Section 6 in our current manuscript for more discussions about the limitation of CuCo.
>
> As required by the reviewer, we ran CuCo on some complex tasks. CuCo achieves superior performance on PlatformJumper and Thrower tasks but fails to beat the baselines on Climber and all methods fail on Lifter, which may require the robot to grow thin limbs or fingers. In the next question, we analyzed in detail the reasons for the failure and the success.

---

> > ### Author Response · Authors · 2022-11-18
> > **Response to Reviewer vM9V (Part 2/2)**
> >
> > >**[Q3]** _By looking at the results, CuCo tends to produce regular shapes for the robot design compared to GA. It does not appear to evolve robots with some thin limb structures. However, my intuition tells me that for some complex tasks (such as climbing/throwing/lifting in environments), thin structures like limbs or fingers are the key to success. Could you please evaluate your method on those environments in Evolution Gym as well and report the performance?_
> >
> > **[A3]** We completely agree with the reviewer’s comment. In Section 5.2, we have additionally evaluated our method and all baselines on these tasks. CuCo learns good throwing skills in Thrower, with the ability to keep the balance and find a better launch angle. Compared with GA and BO, it is true that CuCo fails to grow some irregular structures for solving Climber and Lifter tasks, as noted by the reviewer.
> >
> > **For the failure in Climber and Lifter**
> >
> > We speculate that there are two reasons for the failure of CuCo on these tasks:
> >
> > **Design.** NCA is assigned no design-related reward for fair comparisons and we encourage NCA to learn the connectivity on its own. Specifically, we check the connectivity at each design step and terminate the episode if the resulting design does not meet the connectivity or has no actuator voxels. Therefore, NCA becomes more conservative in using empty voxels. Due to the mechanism of inheritance, this kind of design pattern would be further “exploited” in larger design spaces, restricting the exploration of irregular morphologies.
> >
> > **Control.** We formulate the design and control into a single MDP. For some challenging tasks (e.g., Lifter is hard for all methods), when the control policy is hard to explore and exploit the environmental reward, the design policy will also hardly see the future reward.
> >
> > Fortunately, the failure also suggests that we can (1) add design rewards to encourage NCA to explore some counterintuitive structures and (2) use more complex NCA architectures [1-2] to explore irregular designs. Hence, an important future work is creating a more efficient mechanism for balancing the exploration and exploitation abilities of the NCA-based design policy.
> >
> > In summary, we stay positive about this failure and agree that there is further room for improvement of CuCo.
> >
> > **For the success in Thrower**
> >
> > We also observe that although some irregular morphologies can help accomplish more complex tasks, they are prohibitively expensive to obtain and challenging to control. In other words, we believe that there exists a trade-off between morphology and control [3]. For the Thrower task, it will finally take baseline methods hundreds of millions of simulation steps [4] to achieve a similar performance with CuCo. While GA and BO produce VSRs with arms, they are not strong enough to throw the box further. Instead, CuCo generates beneficial homogeneous tissue structures that can be actuated coordinately by the transformer-based control policy, resulting in smart and powerful VSRs.
> >
> > >**[Q4]** _Could you show the performance/convergence curves for the baseline methods as well?_
> >
> > **[A4]**  Thanks for your suggestion. The learning curves of all methods are shown in the newly added Appendix B.1, and also shown at: https://ibb.co/SN4JpBP.
> >
> >
> > **Reference**
> >
> > [1] Caitlin Grasso and Josh C. Bongard. Empowered neural cellular automata. Proceedings of the
> > Genetic and Evolutionary Computation Conference Companion, 2022.
> >
> > [2] Shyam Sudhakaran, Elias Najarro, and Sebastian Risi. Goal-guided neural cellular automata: Learning to control self-organising systems. ArXiv, abs/2205.06806, 2022.
> >
> > [3] Vincent C. M ̈uller and Matej Hoffmann. What is morphological computation? on how the body contributes to cognition and control. Artificial Life, 23:1–24, 2017.
> >
> > [4] Jagdeep Bhatia, Holly Jackson, Yunsheng Tian, Jie Xu, and Wojciech Matusik. Evolution gym: A large-scale benchmark for evolving soft robots. In NeurIPS, 2021.

---

### Official Review · Reviewer_iFUz · 2022-10-24

**Confidence:** 3
**Correctness:** 3
**Technical Novelty And Significance:** 3
**Empirical Novelty And Significance:** 3
**Recommendation:** 8

**Clarity, Quality, Novelty And Reproducibility:**

### Clarity
A few points in the paper are unclear:
- The paper uses the term "open-loop controller" for describing prior works. However, how exactly this term is used in this setting and how the proposed method differs from an "open-loop controller" is not clear.
- It is not clear whether the morphology is inherited by inheriting the exact previous design or the design policy weights. This seems like a significant distinction since it seems the design policy can be stochastic.
- Related to the previous point, it is not clear whether the control and design policies are deterministic or stochastic.
- Finally, the paper seems to base their architecture and training procedure on Transformr2Act. The authors could make it clearer which aspects of this paper are novel contributions and which are prior work, on which they build.

### Quality
Overall, this paper is well written with clear figures.

### Novelty
This paper validates the effectiveness of pre-defined curriculum over robot size in VSR design. This is a useful, novel contribution that can inform applications and future research in this area, e.g. applying curricula over other parameters of the design space or applying automatic curriculum learning methods to VSR.

### Reproducibility
The algorithm, architecture, and hyperparameters are clearly detailed in this paper, making this work, in principle, reproducible.

**Strength And Weaknesses:**

### Strengths
- The paper is overall well-written with clear figures and diagrams.
- The experimental setting and baselines are clearly communicated and seem appropriate for addressing their research question about whether body-size curricula is useful for co-designing VSR control and morphology.
- The experiments comprehensively ablate each component of the proposed method: the use of a curriculum, the use of self-attention, and whether the morphology or design policy are inherited across curriculum stages.

### Weaknesses
- The study can benefit from an ablation of the choice of using an NCA architecture for the design policy.
- Several key details are left unclear in the current paper, as detailed in the Clarity section.
- A primary weakness with this work is that the results are only validated in simulation. The goal of VSR is to transfer to real robots, and the so the study would be much more impactful if the authors can demonstrate the effectiveness of the learned morphologies and controllers in a sim2real setting.

Lastly, there is important related work around automatic curriculum learning that the authors should cite, especially around unsupervised environment design, which describes the same problem setting that the authors investigate (where the environment is the combination of the task and morphology):
- Matiisen, Tambet, et al. "Teacher–student curriculum learning." IEEE transactions on neural networks and learning systems 31.9 (2019): 3732-3740.
- Portelas, Rémy, et al. "Teacher algorithms for curriculum learning of deep rl in continuously parameterized environments." Conference on Robot Learning. PMLR, 2020.
- Jiang, Minqi, Edward Grefenstette, and Tim Rocktäschel. "Prioritized level replay." International Conference on Machine Learning. PMLR, 2021.
- Dennis, Michael, et al. "Emergent complexity and zero-shot transfer via unsupervised environment design." Advances in neural information processing systems 33 (2020): 13049-13061.
- Jiang, Minqi, et al. "Replay-guided adversarial environment design." Advances in Neural Information Processing Systems 34 (2021): 1884-1897.
- Parker-Holder, Jack, et al. "Evolving Curricula with Regret-Based Environment Design." International Conference on Machine Learning. PMLR, 2022.




**Summary Of The Paper:**

This paper investigates the effectiveness of using a predefined curriculum over the size of a Voxel-Based Soft Robot (VSR) in co-training design and control policies simultaneously using RL (PPO). Their results in training a target 7x7 VSR configuration provides strong evidence that such curricula over VSR body size is effective for producing more performant morphologies and associated control policies on 4 environments in simulation. The architectural and training procedure used in this paper is largely taken from Yuan et al, 2022 (Transform2Act).

**Summary Of The Review:**

Overall, this paper provides a clean investigation of the impact of body-size curricula in co-training morphology design and control policies for VSR. The experiment setting is sound, including several VSR environments, key ablations, and sensible baselines. The main finding—that body-size curricula lead to improved morphologies and control policies in simulation—seems valuable in informing future work in this area.

---

> ### Author Response · Authors · 2022-11-18
> **Response to Reviewer iFUz (Part 1/2)**
>
> We sincerely thank the reviewer for the positive feedback and appreciation of our approach’s novelty and significance. We hope to provide clarification to address the concerned issues. For the convenience of the reviewer, we reiterate the comments below.
>
> >**[Q1]** _The study can benefit from an ablation of the choice of using an NCA architecture for the design policy._
>
> **[A1]** Thanks for your suggestion. We have conducted an additional ablation experiment where we replaced the NCA architecture with a Compositional Pattern Producing Network (CPPN) [1]. In contrast to NCA, CPPN receives the global information of the design space (e.g., $x$ and $y$ coordinates of the design space) and immediately outputs the design parameters. We named this ablation CuCo-CPPN.
>
> The results are shown in Appendix B.2, and also shown at https://ibb.co/L0YvQ6J.
>
> We observe that CuCo-NCA outperforms CuCo-CPPN significantly in terms of convergence speed and final performance due to CPPN doesn’t adapt well to the inheritance mechanism. This ablation further underscores the importance of NCA's advantage to encode practical design patterns and scale them to larger design spaces (morphologies are shown at https://files.catbox.moe/yubr3s.mp4).
>
> >**[Q2]** *A primary weakness with this work is that the results are only validated in simulation. The goal of VSR is to transfer to real robots, and the so the study would be much more impactful if the authors can demonstrate the effectiveness of the learned morphologies and controllers in a sim2real setting.*
>
> **[A2]** We fully agree with the reviewer. We plan to transfer our method to design and control real voxel-based soft robots [2-3]. In our work, we evaluated CuCo on various tasks from the simulator as a proof of concept to show its effectiveness. We believe that CuCo may offer several advantages for the sim2real transfer:
>
> 1. Better sample efficiency compared to population-based methods which are prohibitively expensive;
>
> 2. The transformer-based control policy enables us to train a single controller;
>
> 3. CuCo supports end-to-end training of both design and control policies.
>
> It’s true that there are many endeavors to be done to overcome the reality gap, we hope that our work can serve as a step towards realizing the significance of curriculum learning in the field of robot co-design.
>
> >**[Q3]** *Lastly, there is important related work around automatic curriculum learning that the authors should cite, especially around unsupervised environment design, which describes the same problem setting that the authors investigate (where the environment is the combination of the task and morphology)*
>
> **[A3]** We sincerely thank the reviewer for providing valuable references. We have carefully studied these papers and included the citation accordingly in the second paragraph of the Related Work section.
>
> >**[Q4]** *The paper uses the term “open-loop controller” for describing prior works. However, how exactly this term is used in this setting and how the proposed method differs from an “open-loop controller” is not clear.*
>
> **[A4]** Here, the "open-loop controller" parameterizes the control as a simple periodic sequence of actuation that depends only on the time, represented as $f(t)$. Sinusoidal signals, for instance, can be used to vary the volume of voxels periodically, thus, $−1$ corresponds to the maximum requested expansion and $1$ corresponds to the maximum requested contraction [4].
>
> Different from prior works which seldom involve the learning of control, we use the neural network as the VSR’s controller, which maps the current state to the output control signals for actuator voxels, represented as $f(state, t)$.  We train it using reinforcement learning, which enables robots to learn complex non-periodic tasks such as throwing a box, walking on uneven terrains. By contrast, open-loop controllers based on the pre-defined primitives are incapable of these tasks.
>
>  We have made it clearer in the revision Section 2.
>
> **Reference**
>
> [1] Nick Cheney, Robert MacCurdy, Jeff Clune, and Hod Lipson. Unshackling evolution: evolving soft robots with multiple materials and a powerful generative encoding. In GECCO ’13, 2013.
>
> [2] Sam Kriegman, S. Walker, Dylan S. Shah, Michael Levin, Rebecca Kramer-Bottiglio, and Josh C. Bongard. Automated shapeshifting for function recovery in damaged robots. ArXiv, abs/1905.09264, 2019.
>
> [3] Sam Kriegman, Amir Mohammadi Nasab, Dylan S. Shah, Hannah Steele, Gabrielle Branin, Michael Levin, Josh C. Bongard, and Rebecca Kramer-Bottiglio. Scalable sim-to-real transfer of soft robot designs. 2020 3rd IEEE International Conference on Soft Robotics (RoboSoft), pp. 359 -366, 2020.
>
> [4] Jacopo Talamini, Eric Medvet, and Stefano Nichele. Criticality-driven evolution of adaptable mor-phologies of voxel-based soft-robots. Frontiers in Robotics and AI, 8, 2021

---

> > ### Author Response · Authors · 2022-11-18
> > **Response to Reviewer iFUz (Part 2/2)**
> >
> > >**[Q5]** *It is not clear whether the morphology is inherited by inheriting the exact previous design or the design policy weights. This seems like a significant distinction since it seems the design policy can be stochastic.*
> >
> > **[A5]** We apologize for the confusion. In CuCo, both the converged morphology and the weights of the design policy will be inherited from the last learning stage.
> >
> > The inherited morphology will serve as the new initial seed, which will be placed at the center of the design space as the initial state of each episode, and the inherited design policy will work on this seed to leverage the learned design patterns from the last stage to grow larger bodies.
> >
> > Two ablations **CuCo-NIM** (No inheritance of the converged morphology) and **CuCo-NID** (No inheritance of the design policy) show the significance of these two operations (Section5.3).
> >
> > We have made this statement clearer in the revision Section 4.1.
> > >**[Q6]** *It is not clear whether the control and design policies are deterministic or stochastic.*
> >
> > **[A6]** Both the design and control policies of CuCo are stochastic during training and deterministic during evaluation.
> >
> > **Design policy.** It maps the input to a categorical action distribution created by the output logits. During training, the new cell state is determined by sampling from the distribution. While in evaluation, the design policy selects the action that corresponds to the highest logit value.
> >
> > **Control policy.** It outputs the mean of the continuous action, and we use a constant standard deviation to create the Gaussian distribution. During training, the action is randomly sampled from the Gaussian action distribution. While evaluating the agent, only the mean action is exploited for control.
> >
> > We have clarified these 2 issues accordingly in Section 4.1 and Section 4.2, respectively.
> >
> > >**[Q7]** *The paper seems to base their architecture and training procedure on Transformr2Act. The authors could make it clearer which aspects of this paper are novel contributions and which are prior work, on which they build.*
> >
> > **[A7]** Thank you for the suggestion. We have made the statement clearer in the Related Work section.
> >
> > **CuCo draws inspiration from Transform2Act in :**
> >
> > Unifying the design and control in a single decision-making process (MDP), and using reinforcement learning to simultaneously train the design and control policies.
> >
> > **CuCo differs from Transform2Act in :**
> >
> > 1. **Curriculum (CL vs Learning in the fixed target design space).** In CuCo, we expand the modular robot design space from a small size to the target size gradually through a predefined curriculum  (e.g., $3 \times 3 \to 5\times 5 \to 7 \times 7$), while the Transform2Act is applied in a fixed rigid-bodied robot design space.
> >
> > 2. **Design policy (NCA vs GNN).** VSRs can be viewed as "pictures", where each voxel is surrounded by its neighboring voxels. NCA learns a parameterized "convolution kernel " (as shown in Figure 2), which captures the connectivity of the robot's morphology automatically based only on local interactions. Hence, we use NCA as the design policy to encode design patterns for voxels, which enables us to freely scale them across learning stages. In contrast, Transform2Act used GNN. The connectivity of the robot's morphology needs to be represented explicitly for the learning of GNN, which may not be efficient in our VSR setting. The reason is the complexity of modeling the morphology would increase greatly when the dimension of design space is expanded (e.g., when the design is expanded from $5\times 5 \to 7 \times 7$).
> >
> > 3. **Control policy (Transformer vs GNN).** GNNs use morphological information to define the message-passing scheme. Therefore, they need to learn multi-hop communications between modules. However, as the number of modules increases, crucial state information of modules can be difficult to aggregate across multiple hops. Instead, CuCo utilizes Transformer as the control policy, which dynamically calculates the self-attention coefficient among modules through a single pass. Thus, Transformer doesn't need to learn multi-hop communications, which may be more efficient for module robot systems. Due to the self-attention mechanism, it is less affected by the interchange of modules, enabling VSRs to be disassembled and reassembled with more flexibility.

---

### Official Review · Reviewer_Qe3n · 2022-10-26

**Confidence:** 4
**Correctness:** 3
**Technical Novelty And Significance:** 3
**Empirical Novelty And Significance:** 3
**Recommendation:** 6

**Clarity, Quality, Novelty And Reproducibility:**

This paper introduces a new solution for co-design of morphology and control for soft robots, and originally addresses several key challenges. The paper is well written, and the ideas are clearly justified. The experimental results could be reproduced using the Evolution Gym simulator.

**Strength And Weaknesses:**

Strength:

+ The paper addresses an important problem on co-design of morphology and control with the potential to greatly impact voxel-based soft robots.

+ The paper identifies several interesting challenges for soft robot co-design, such as the high dimensionality of the joint design and control space and the difficulty in generalizability. Methods are designed to address these specific challenges.

+ The paper is well organized and well written.

Weakness:

- The goal of the paper is the co-design of morphology and control. However, the approach assumes that "The morphology of the robot is unchangeable during the interaction with the environment". While the control policy uses the learned morphology, but how does the control inform or help morphology design?

- In the related work section, the paper argues that methods such as Transform2Act by Yuan et al (2022) have "the limitation of aggregating multi-hop information". In the experiment section, the paper states that CuCo-NCU (CuCo without the curriculum component) is similar to Transform2Act. How is this multi-hop information aggregated by CuCo. Other than the curriculum, what are the key differences between CuCo and Transform2Act?

- While simulators such as Evolution Gym and MuJoCo are helpful to evaluate theories, out of curiosity, how are these simulators, the proposed method, and the experimental results applicable to real soft robots? For example, what are the existing prototypes of real voxel-based soft robots that can use the proposed method for co-design?


**Summary Of The Paper:**

This paper introduces a new curriculum-based method for co-designing morphology and control of voxel-based soft robots. This curriculum-based method expands the design space from a small size to the target size using reinforcement learning with a predefined curriculum. To address incompatible state-action spaces, local observations of robot voxels are modeled as a sequence and self-attention is used to control the voxels.

**Summary Of The Review:**

Although several technical details need to be clarified, this paper proposes a new method and addresses several specific challenges for morphology and control of simulated voxel-based soft robots.

---

> ### Author Response · Authors · 2022-11-18
> **Response to Reviewer Qe3n (Part 1/2)**
>
> We are glad that the reviewer characterized our work as “interesting” and “important”. We thank the reviewer for the useful and detailed comments. In the following, we aim to address your questions and concerns. For the convenience of the reviewer, we reiterate the comments below.
>
> >**[Q1]** _The goal of the paper is the co-design of morphology and control. However, the approach assumes that “The morphology of the robot is unchangeable during the interaction with the environment”. While the control policy uses the learned morphology, but how does the control inform or help morphology design?_
>
> **[A1]** We acknowledge that this statement is a bit confusing. The word "unchangeable" in this context means when a designed robot is presented in the simulator (e.g., Evolution Gym), the morphology is kept fixed during the simulation. In the meantime, the simulator doesn’t allow us to change the robot morphology once the simulation starts.
>
> Specifically, within each learning stage, we unify the design and control processes into a single decision-making process (MDP). For each episode, the design policy first takes a series of design actions for a few steps to develop a robot, during which the morphology is changeable. Afterwards, the designed robot is controlled and simulated in the simulator for the remaining steps, during which the morphology is unchangeable.
>
> Recall that we unify both the design and control processes into a single MDP. Although we don’t assign any rewards (we set the reward to zero) to the design policy during design steps, it will see future rewards collected by the control policy under the designed morphology, which provides learning signals for the design policy. It also explains why the morphology is unchangeable during the simulation (to generate a complete trajectory of MDP for training). In addition, both the design and control policies share the same value function and are jointly updated to maximize the predicted value using the policy gradient approach.
>
> We have clarified it accordingly in Section 4, thanks.
>
> >**[Q2]** _In the related work section, the paper argues that methods such as Transform2Act by Yuan et al (2022) have “the limitation of aggregating multi-hop information”. In the experiment section, the paper states that CuCo-NCU (CuCo without the curriculum component) is similar to Transform2Act. How is this multi-hop information aggregated by CuCo?_
>
> **[A2]** We would like to clarify that in CuCo and CuCo-NCU, the problem of aggregating multi-hop information is handled by the transformer-based control policy.
>
> CuCo-NCU has the same architectures of the design and control policies (NCA and Transformer) as CuCo but without the curriculum learning, that is, it learns to design and control in a fixed target design space (e.g., $7 \times 7$), and the Transform2Act [1] also learns to design and control in a fixed target design space.
>
> We have clarified it accordingly in Section 5.1.
>
> Note that the control of modular robots is related to handling relationships between modules' state information, which is the key to designing the controller for a modular robot. GNNs use morphological information to define the message-passing scheme, thus, they need to learn multi-hop communications between modules [2-3]. However, Transformer (CuCo's control policy), which can be characterized as a fully connected GNN with attentional aggregation [4-5], handles the relationships between modules by dynamically calculating the self-attention coefficient. Thus, Transformer doesn't need to learn multi-hop communications.
>
> **Reference**
>
> [1] Ye Yuan, Yuda Song, Zhengyi Luo, Wen Sun, and Kris M. Kitani. Transform2act: Learning a transform-and-control policy for efficient agent design. ArXiv, abs/2110.03659, 2022.13.
>
> [2] Wenlong Huang, Igor Mordatch, and Deepak Pathak. One policy to control them all: Shared modular
> policies for agent-agnostic control. In ICML, 2020.
>
> [3] Federico Pigozzi, Yujin Tang, Eric Medvet, and David Ha. Evolving modular soft robots without explicit inter-module communication using local self-attention. Proceedings of the Genetic and Evolutionary Computation Conference, 2022.
>
> [4] Vitaly Kurin, Maximilian Igl, Tim Rockt ̈aschel, Wendelin Boehmer, and Shimon Whiteson. My body is a cage: the role of morphology in graph-based incompatible control. ArXiv, abs/2010.01856, 2021.
>
> [5] Agrim Gupta, Linxi (Jim) Fan, Surya Ganguli, and Li Fei-Fei. Metamorph: Learning universal controllers with transformers. ArXiv, abs/2203.11931, 2022.

---

> > ### Author Response · Authors · 2022-11-18
> > **Response to Reviewer Qe3n (Part 2/2)**
> >
> > >**[Q3]** _What are the key differences between CuCo and Transform2Act?_
> >
> > **[A3]** CuCo differs from Transform2Act in :
> >
> > 1. **Curriculum (CL vs Learning in the fixed target design space).** In CuCo, we expand the modular robot design space from a small size to the target size gradually through a predefined curriculum (e.g., $3 \times 3 \to 5\times 5 \to 7 \times 7$), while Transform2Act is applied in a fixed rigid-bodied robot design space.
> >
> > 2. **Design policy (NCA vs GNN).** VSRs can be viewed as "pictures", where each voxel is surrounded by its neighboring voxels. NCA learns a parameterized "convolution kernel " (as shown in Figure 2), which captures the connectivity of the robot's morphology automatically based only on local interactions. Hence, we use NCA as the design policy to encode design patterns for voxels, which enables us to freely scale them across learning stages. In contrast, Transform2Act used GNN. The connectivity of the robot's morphology needs to be represented explicitly for the learning of GNN, which may not be efficient in our VSR setting. The reason is the complexity of modeling the morphology would increase greatly when the dimension of design space is expanded (e.g., when the design is expanded from $5\times 5 \to 7 \times 7$).
> >
> > 3. **Control policy (Transformer vs GNN).** GNNs use morphological information to define the message-passing scheme. Therefore, they need to learn multi-hop communications between modules. However, as the number of modules increases, crucial state information of modules can be difficult to aggregate across multiple hops. Instead, CuCo utilizes Transformer as the control policy, which dynamically calculates the self-attention coefficient among modules through a single pass. Thus, Transformer doesn't need to learn multi-hop communications, which may be more efficient for module robot systems. Due to the self-attention mechanism, it is less affected by the interchange of modules, enabling VSRs to be disassembled and reassembled with more flexibility.
> >
> > >**[Q4]** _While simulators such as Evolution Gym and MuJoCo are helpful to evaluate theories, out of curiosity, how are these simulators, the proposed method, and the experimental results applicable to real soft robots? For example, what are the existing prototypes of real voxel-based soft robots that can use the proposed method for co-design?_
> >
> > **[A4]** We fully agree with the reviewer that this is a very promising research direction. In our work, we evaluated CuCo on various tasks from Evolution Gym as a proof of concept to show its effectiveness. We believe that the physical VSRs (e.g., the Voxcraft platform) proposed in the literature [1-2] can bring insights for transferring CuCo to reality.
> >
> > For instance, multi-material cubic blocks fabricated by 3D printing can be used to build the VSR’s morphology, and each voxel can be equipped with several sensors (e.g., touch, pressure, and velocity sensors) to form the local observation space. Moreover, the volumetric actuation can be adapted to the neural network controller.
> >
> > We believe that CuCo may offer several advantages for the sim2real transfer:
> >
> > 1.Better sample efficiency compared to population-based methods which are prohibitively expensive;
> >
> > 2.The transformer-based control policy enables us to train a single controller;
> >
> > 3.CuCo supports end-to-end training of both design and control policies.
> >
> > Our work lies in the intersection of robotics, artificial life, curriculum learning and reinforcement learning. Although it is a small step towards efficient co-design of VSRs, we believe it will inspire future research on automatic co-design of physical modular soft robots.
> >
> >
> > Thank you for your review and please let us know if you have any additional comments, questions, or concerns.
> >
> > **Reference**
> >
> > [1] Sam Kriegman, S. Walker, Dylan S. Shah, Michael Levin, Rebecca Kramer-Bottiglio, and Josh C. Bongard. Automated shapeshifting for function recovery in damaged robots. ArXiv, abs/1905.09264, 2019.
> >
> > [2] Sam Kriegman, Amir Mohammadi Nasab, Dylan S. Shah, Hannah Steele, Gabrielle Branin, Michael Levin, Josh C. Bongard, and Rebecca Kramer-Bottiglio. Scalable sim-to-real transfer of soft robot designs. 2020 3rd IEEE International Conference on Soft Robotics (RoboSoft), pp. 359 -366, 2020.

---

### Author Response · Authors · 2022-11-18
**Summary of response**

We sincerely thank all the reviewers for their valuable and detailed feedback. We are glad that the reviewers found: (1) Our research problem is interesting and significant (reviewer Qe3n). (2) Our proposed method CuCo is a novel and sound approach with the potential to impact voxel-based soft robots (all reviewers). (3) Our experimental studies and analysis are clear, comprehensive and informative, providing impressive visual behavior and empirical performance (reviewer iFUz, vM9V).

Based on the reviewers’ comments we (A) revised the paper and clarified some unclear and possibly confusing expressions, and (B) conducted additional experiments to cover more tasks and performance analysis. Here, we list the main changes:

&ensp; **Detailed analysis of our approach**

&ensp;1. We clarified several technical details of our design and control policies in Section 4 (reviewer Qe3n, iFUz);

&ensp;2. We clarified the difference between our approach and research such as Transform2Act (reviewer Qe3n, iFUz) in Section 2;

&ensp;3. We added more discussions about the current limitation of CuCo in Section 6 (reviewer vM9V).

&ensp; **More experiments on different tasks**

&ensp;1. Inspired by reviewer iFUz, we provided an ablation of the choice of using NCA as the design policy in Appendix B.2;

&ensp;2. We evaluated CuCo on 4 additional tasks (reviewer vM9V). Results in Section 5.2 show that our approach outperforms baselines on most tasks, and we also discussed in detail the failure cases.

The video of supplementary results is available here: https://files.catbox.moe/yubr3s.mp4.

Please let us know if there are any additional comments, questions, or concerns. Many thanks again for the invaluable hard work of all reviewers to improve our work.

---

> ### Comment · Reviewer_iFUz · 2022-11-19
> **Response to authors**
>
> My opinion of the paper has not changed in light of the authors' response. I will keep my current rating for this paper.

---

### Decision · Program_Chairs · 2023-01-20

**Decision:**

Accept: poster

**Justification For Why Not Higher Score:**

Scope of this paper is limited to soft robot design and control. This may not be very relevant to the broad ICLR audience.

**Justification For Why Not Lower Score:**

Strong technical ideas and strong experimental results.

**Metareview: Summary, Strengths And Weaknesses:**

This paper presents a new co-designing curriculum for optimizing the morphology of a soft robot along with its control. The reviewers unanimously agree that the ideas presented here are quite interesting and impactful for agent design. There are several other technical ideas such as transforming the observations of the robot as a sequence compatible with self-attention. The only concern here is the relevance of voxel-based soft-robot design in simulation for the ICLR community.

**Note From Pc:**

if the above contains the word "oral" or "spotlight" please see: "oral" presentation means -> notable-top-5% and "spotlight" means -> notable-top-25%. As stated in our emails, we are disassociating presentation type from AC recommendations